# A DISTINCT UNSUPERVISED REFERENCE MODEL FROM THE ENVIRONMENT HELPS CONTINUAL LEARNING

## ABSTRACT

The existing continual learning methods are mainly focused on fully-supervised scenarios and are still not able to take advantage of unlabeled data available in the environment. Some recent works tried to investigate semi-supervised continual learning (SSCL) settings in which the unlabeled data are available, but it is only from the same distribution as the labeled data. This assumption is still not general enough for real-world applications and restricts the utilization of unsupervised data. In this work, we introduce **Open-Set Semi-Supervised Continual Learning (OSSCL)**, a more realistic semi-supervised continual learning setting in which out-of-distribution (OoD) unlabeled samples in the environment are assumed to coexist with the in-distribution ones. Under this configuration, we present a model with two distinct parts: (i) the reference network captures general-purpose and task-agnostic knowledge in the environment by using a broad spectrum of unlabeled samples, (ii) the learner network is designed to learn task-specific representations by exploiting supervised samples. The reference model both provides a pivotal representation space and also segregates unlabeled data to exploit them more efficiently. By performing a diverse range of experiments, we show the superior performance of our model compared with other competitors and prove the effectiveness of each component of the proposed model.

## 1 INTRODUCTION

In a real-world continual learning (CL) problem, the agent has to learn from a non-i.i.d. stream of samples with serious restrictions on storing data. In this case, the agent must be prone to catastrophic forgetting during training (French, 1999). The existing CL methods are mainly focused on supervised scenarios and can be categorized into three main approaches (Parisi et al., 2019): (i) Replay-based methods reuse samples from previous tasks either by keeping raw samples in a limited memory buffer (Rebuffi et al., 2017; Lopez-Paz & Ranzato, 2017; Aljundi et al., 2019) or by generating pseudo-samples from previous classes (Shin et al., 2017; Wu et al., 2018; van de Ven et al., 2020). (ii) Regularization-based methods aim to maintain the stability of the network across tasks by penalizing deviation from the previously learned representations or parameters (Nguyen et al., 2018; Cha et al., 2021; Rebuffi et al., 2017; Li & Hoiem, 2016). (iii) Methods based on parameter isolation dedicate distinct parameters to each task by introducing new task-specific weights or masks (Rusu et al., 2016; Yoon et al., 2018; Wortsman et al., 2020).

Humans, as intelligent agents, are constantly in contact with tons of unsupervised data being endlessly streamed in the environment that can be used to facilitate concept learning in the brain (Zhuang et al., 2021; Bi & Poo, 1998; Hinton & Sejnowski, 1999). With this in mind, an important but less explored issue in many practical CL applications is how to effectively utilize a vast stream of unlabeled data along with limited labeled samples. Recently, efforts have been made in this direction leading to the investigation of three different configurations: Wang et al. (2021) introduced a very restricted scenario for semi-supervised continual learning in which the unsupervised data are only from the classes which are being learned at the current time step. On the other hand, Lee et al. (2019) introduced a configuration that is "more similar to self-taught learning rather than semi-supervised learning". In fact, they introduced a setting in which the model is exposed to plenty of labeled samples which is a necessary assumption for their model to achieve a good performance; in addition, their model has access to a large corpse of unsupervised data in an environment that typically does not include samples related to the current CL problem. By adopting this idea, Smith et al. (2021) proposed a

more realistic setting by assuming a limitation on the number of supervised samples available for the training. In addition to that, they assumed the existence of a shared hidden hierarchy between the supervised and unsupervised samples, which is not necessarily true for practical applications.

In this work, we will first propose a general scenario to unify the mentioned configurations into a more realistic setting called **Open-Set Semi-Supervised Continual Learning** (OSSCL). In this scenario, the agent can observe unsupervised data from two sources: (i) Related unsupervised data, which are sampled from the same distribution as the supervised dataset, and (ii) Unrelated unsupervised data which have a different distribution from the classes of the current CL problem. The in-distribution unsupervised samples can be from the classes that are being solved, have been solved at previous time steps, or are going to be solved in the future.

Previous CL works in which unlabeled data was available alongside labeled data, mainly utilized unlabeled data by creating pseudo-labels for them using a model which is trained by labeled samples (Lee et al., 2019; Smith et al., 2021; Wang et al., 2021). Those unlabeled data with their pseudo-labels were used directly in the training procedure. However, due to the fact that labeled data are scarce in realistic scenarios, the pseudo-labeling process will be inaccurate and creates highly noisy labels. Therefore, we present a novel method to learn in the OSSCL setting which alleviates the mentioned problem and utilizes unlabeled data effectively.

Our proposed model, which is consisted of an Unsupervised Reference network and a Supervised Learner network (URSL), can effectively absorb information by leveraging contrastive learning techniques combined with knowledge distillation methods in the representation space. While the reference network is mainly responsible for learning general knowledge from unlabeled data, the learner network is expected to capture task-specific information from a few supervised samples using a contrastive loss function. In addition, the learner retains a close connection to the reference network to utilize the essential related information provided by unsupervised samples. At the same time, the representation space learned in the reference network can be utilized to provide an out-of-distribution detector that segregates unlabeled data to employ the filtered ones more properly in the training procedure of the learner model. In short, our main contributions are as follows:

- We propose OSSCL as a realistic semi-supervised continual learning scenario that an intelligent agent encounters in practical applications (Section 2).

- We propose a novel dual-structured model that is suitable for learning in the mentioned scenario and can effectively exploit unlabeled samples (Section 3).

- We show the superiority of our method in several benchmarks and different combinations of unlabeled samples. our model achieves state-of-the-art accuracy with a notable gap compared to the baselines and previous methods (Section 4).

## 2 Preliminaries

In this work, we consider the training dataset to consist of two parts; the supervised dataset $\mathcal{D}_{sup}$ is a sequence of $T$ tasks $\{\mathcal{T}_1, \mathcal{T}_2, ..., \mathcal{T}_T\}$. At time step $t$, the model only has access to $\mathcal{T}_t = \{(x_i, y_i)\}_{i=1}^{N_t}$ where $x_i \overset{i.i.d.}{\sim} P(X|y_i)$ denotes a training sample and $y_i$ represents its corresponding label. We consider $K$ separate classes at each task and follow the common class-incremental setting as it is shown to be the most challenging scenario for evaluation. Given a training loss $\ell$ and the network parameters $\theta$, the training objective at time step $t$ is defined as $\theta^* = \arg\min_\theta \frac{1}{N_t} \sum_{i=1}^{N_t} \ell(x_i, y_i, \theta)$.

On the other hand, the unsupervised dataset $\mathcal{D}_{unsup}$ is a sequence of $T$ sets $\{\mathcal{U}_1, \mathcal{U}_2, ..., \mathcal{U}_T\}$ containing only unlabeled data points. We assume that $\mathcal{U}_t$ represents the unsupervised data available in the environment at time step $t$, which is accessible by the model along with $\mathcal{T}_t$. Based on the OSSCL setting which is a general framework, we assume that the unsupervised dataset is composed of two parts: (i) The related part, also called the in-distribution set, is consisted of unsupervised samples generated from the same distribution as $\mathcal{D}_{sup}$. In order to maintain generality, we assume that this set consists not only of unsupervised samples related to the current supervised task but also of the other tasks of the CL problem that have either been observed in previous time steps or will be observed in the future. (ii) The unrelated data points, also called the out-of-distribution samples, are a set of unsupervised data sampled from the distribution $Q$, which is not necessarily the distribution from

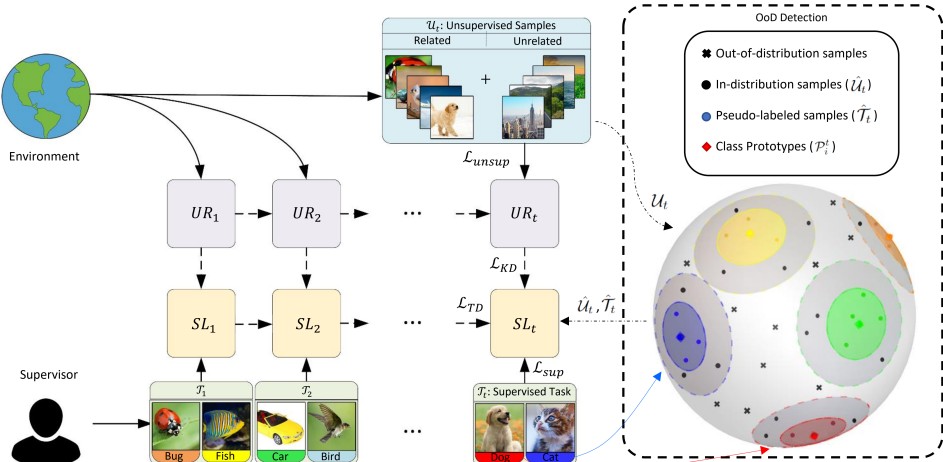

Figure 1: A schematic of the method and configuration. The unsupervised reference ($UR_t$), supervised learner ($SL_t$), labeled data ($\mathcal{T}_t$), and related and unrelated unlabeled data ($\mathcal{U}_t$) at time step $t$ are shown on the left while the OoD segregation module is shown on the right of the figure.

which the supervised samples have been generated. In the next section, we will propose a novel method to perform in this configuration, and in Section 4, a variety of experiments are provided to show the effectiveness of our model.

## 3 METHOD

Learning continually from $\mathcal{D}_{sup}$ has been widely explored by the community. Meanwhile, unlike deep models, humans are less hungry for supervised data. Although they observe a large volume of data during their lifetime, only a small and insignificant portion of this data is labeled. It is believed that the considerable human ability to learn with a few instances is due to the rich representations learned from the large volumes of unsupervised observations (Zhuang et al., 2021; Bi & Poo, 1998; Hinton & Sejnowski, 1999). Here, we aim to explore the benefits of using $\mathcal{D}_{unsup}$ and its impact on empowering the continual learner. Specifically, we will show how $\mathcal{D}_{unsup}$ will promote representation learning in addition to providing positive forward/backward transfer in the continual learning process.

We propose our URSL model, which is consisted of two parts: 1) The general task-agnostic reference network, which is responsible for absorbing information from unsupervised data in the environment, and 2) the learner network, which is designed to capture knowledge from a few supervised samples while it is also guided by the reference network. The notation $UR_t$ and $SL_t$ are used to respectively demonstrate the reference and learner network instances at time step $t$ (refer to Figure 1 and Algorithm 1 for an overview).

We employ a contrastive representation learning approach for training both the reference and the learner networks. This approach has been proven to be a proper solution for supervised CL problems. Indeed, some previous works in CL claimed that classifier heads placed on top of the representation network are the serious sources of catastrophic forgetting (Ramasesh et al., 2021; Banayeeanzade et al., 2021; Cha et al., 2021), therefore, Co$^2$L (Cha et al., 2021) presented a supervised contrastive loss to avoid this problem. We utilize contrastive representation learning as a unified approach for training both the reference and the learner networks, which allows information to flow between these networks easily. Combined with knowledge distillation techniques applied in the representation space, this approach provides a convenient tool to exploit the most out of unsupervised samples.

Our model is also equipped with an exemplar memory $\mathcal{M}$ to randomly store a portion of supervised samples from previous tasks (Lopez-Paz & Ranzato, 2017; Rebuffi et al., 2017). The stored samples will contribute to the training of the learner network. After the final time step, these samples are also used to train a classifier head on top of the representation space of the learner network. It is noteworthy that our model does not store unlabeled data in its own memory since this data is always found in abundance in the environment, and this makes our model needless of a large memory.

## 3.1 Reference Network

The unsupervised reference network $UR_t : \mathcal{X} \to \mathbb{R}^d$ is a general-purpose feature extractor responsible for encoding all kinds of unsupervised information available in the environment. The network is composed of an encoder $f$, and a projector $g$, responsible for embedding input $x$ in the representation space by $z = (f \circ g)_{\theta_t}(x)$ where $z$ is on the unit $d$-dimensional Euclidean sphere and $\theta_t$ represents the model parameters at time step $t$. Considering a batch $B \subseteq \mathcal{U}_t$ with size $N$, the SimCLR (Chen et al., 2020a) loss function used for training the network can be written as:

$$
h_{i,j} = -\log \frac{\exp(\tilde{z}_i . \tilde{z}_j / \tau)}{\sum\limits_{k=1}^{2N} \mathbb{1}_{[k \neq i]} \exp(\tilde{z}_i . \tilde{z}_k / \tau)}, \quad \mathcal{L}_{unsup}(\theta_t; \tau) = \frac{1}{2N} \sum_{k=1}^{N} (h_{2k-1,2k} + h_{2k,2k-1}), \quad (1)
$$

where $\tilde{z}_{2i}$ and $\tilde{z}_{2i-1}$ are the representations of two different augmentations of the same image $x_i \in B$ and $\tau$ is the temperature hyperparameter.

## 3.2 Segregating Unsupervised Samples

In this section, we show how to segregate unlabeled samples by employing the reference network and supervised samples. Although unsupervised samples can play an important role in both learning the representation space and controlling changes in this space through time, naive approaches to incorporating these samples into the training of the learner network can lead to inferior performance due to the existence of unrelated samples among unlabeled ones. Therefore, we will first explain the OoD detection method, which is designed to segregate unlabeled data and incorporate them more properly in the continual learning process of the learner network. To efficiently segregate unsupervised data, we employ a prototypical-based OoD detection method (Park et al., 2021) in the representation space of the reference network using samples in $\mathcal{T}_t \cup \mathcal{M}$. It is noteworthy that the representation space of the reference network is chosen for OoD detection since it provides better sample discrimination than any other representation space obtained by training over a small number of labeled samples. Additionally, this approach eliminates the need to train another network specialized in OoD detection in contrast to the previous works (Chen et al., 2020b; Huang et al., 2021; Saito et al., 2021).

At time step $t$, our OoD method creates $\mathcal{P}^t = \left\{ \mathcal{P}^t_1, \mathcal{P}^t_2, \ldots, \mathcal{P}^t_{K \times t} \right\}$, a set of $K \times t$ prototypes representing the centroids of observed classes so far, which is extracted using the labeled data available in $\mathcal{T}_t \cup \mathcal{M}$:

$$
\mathcal{P}^t_i = \psi \left( \frac{1}{|\mathcal{A}| \sum\limits_{(x_j, y_j) \in \mathcal{T}_t \cup \mathcal{M}} \mathbb{1}_{[y_j = i]}} \sum\limits_{(x_j, y_j) \in \mathcal{T}_t \cup \mathcal{M}} \mathbb{1}_{[y_j = i]} \sum_{a \in \mathcal{A}} (f \circ g)_{\theta_t}(a(x_j)) \right), \quad (2)
$$

where $\mathcal{A}$ is a set of augmentations meant to form different views of a real image, and $\psi$ is the operator that projects vectors into the unit $d$-dimensional sphere. We also define the score operator $\mathcal{S}(\mathcal{P}^t, z) = \max\limits_i \ c(\mathcal{P}^t_i, z)$ where $c$ denotes the cosine similarity measure. This operator takes prototypes in addition to a sample in the representation space and calculates the score of its most probable assignment. With this in mind, we consider $S^t_l$ as the scores of the labeled data obtained by passing $\mathcal{T}_t \cup \mathcal{M}$ through the $\mathcal{S}(\mathcal{P}^t, .)$ operator, i.e. : $S^t_l = \{\mathcal{S}(\mathcal{P}^t, (f \circ g)_{\theta_t}(x)) | x \in \mathcal{T}_t \cup \mathcal{M}\}$. By considering $\eta_{id}$ as a hyperparameter, we define a threshold $\tau_{id} = \text{mean}(S^t_l) + \eta_{id}\text{var}(S^t_l)$ on the scores of unlabeled data to specify in-distribution samples as $\hat{\mathcal{U}}_t = \{x | x \in \mathcal{U}_t, \mathcal{S}(\mathcal{P}^t, x) > \tau_{id}\}$. Furthermore, we assign pseudo-labels to the unsupervised samples on which we have superior confidence by defining a higher threshold $\tau_{pl} = \text{mean}(S^t_l) + \eta_{pl}\text{var}(S^t_l)$, with the hyperparameter $\eta_{pl}$, and prepare pseudo-labeled samples as $\hat{\mathcal{T}}_t = \{(x, \hat{y}) | x \in \mathcal{U}_t, \mathcal{S}(\mathcal{P}^t, x) > \tau_{pl}, \hat{y} = \arg\max_i c(\mathcal{P}^t_i, x)\}$. In other words, an unsupervised sample with a similarity value higher than $\tau_{pl}$ to a class prototype is pseudo-labeled to that class. However, to reduce pseudo-labeling noise, we do not utilize pseudo-labels directly during the training procedure. Those pseudo-labels are used to identify whether this unlabeled data is from past classes or not. Samples of $\hat{\mathcal{T}}_t$ are mainly used to compensate for the small number of supervised samples in the memory, as further explained in the next section. We provide a detailed investigation of the performance of the OoD module in Appendix C.

### 3.3 LEARNER NETWORK

Similar to the reference network, the learner network $SL_t : \mathcal{X} \to \mathbb{R}^d$ is a feature extractor with the form $z = (f \circ g)_{\varphi_t}(x)$ where $\varphi_t$ denotes the model parameters at time step $t$. The training of the learner network is done using three mechanisms:

**Supervised Training:** Following Co²L (Cha et al., 2021), we will use an asymmetric supervised version of the contrastive loss function to train the learner network. By considering a supervised batch $B = \{(x_i, y_i)\}_{i=1}^N$, which is sampled from $\mathcal{T}_t \cup \mathcal{M} \cup \hat{\mathcal{T}}_t$, and applying an augmentation policy to form two different views of real samples, we can write the supervised contrastive loss as follow:

$$\mathcal{L}_{\text{sup}}(\varphi_t; \tau) = \frac{1}{N} \sum_{i=1}^N \frac{-\mathbb{1}_{[y_i \in O_t]}}{|\zeta_i|} \sum_{j \in \zeta_i} \log \frac{\exp(\tilde{z}_i . \tilde{z}_j / \tau)}{\sum\limits_{k=1}^N \mathbb{1}_{[k \neq i]} \exp(\tilde{z}_i . \tilde{z}_k / \tau)}, \tag{3}$$

where $O_t$ is the new classes of the current time step $t$, and $\zeta_i$ are the other samples of the current batch with the same label $y_i$.

The existence of $\hat{\mathcal{T}}_t$ is crucial for learning a proper representation since only a small amount of labeled data is available during continual learning. In fact, Co²L intends to prevent overfitting to the small number of past task samples stored in the memory by proposing the asymmetric supervised contrastive loss that utilizes samples from the memory only as negative samples (Cha et al., 2021). However, when the labeled data are limited, even employing the past samples in $\mathcal{M}$, as negative samples, still may cause overfitting. Therefore, we enrich $\mathcal{M}$ by $\hat{\mathcal{T}}_t$ to diversify the samples from previous classes.

**Knowledge Transfer Through Time:** The loss function in Eq. 3 allows the model to discriminate between new and previous classes. However, it is not sufficient to maintain the discrimination power of the learner network among previous tasks. Therefore to avoid catastrophic forgetting, at each time step $t$, we use an instance-wise relation distillation (IRD) loss to transfer knowledge from the previous time step to the current model (Cha et al., 2021). This self-distillation technique, which is also compatible with the contrastive representation learning approach, retains the old knowledge by maintaining the samples' similarity in the representation space of the learner network. To this end, first, we sample a batch $B$ from $\mathcal{T}_t \cup \mathcal{M} \cup \hat{\mathcal{T}}_t$, augment each sample $x_i$ twice to create $\tilde{x}_{2i-1}, \tilde{x}_{2i}$, and then calculate the instance-wise similarity vector as:

$$p(\tilde{x}_i; \varphi, \tau) = [p_{i,0}, \dots, p_{i,i-1}, p_{i,i+1}, \dots, p_{i,2N}] \text{ where } p_{i,j} = \frac{\exp(\tilde{z}_i . \tilde{z}_j / \tau)}{\sum\limits_{k=1}^{2N} \mathbb{1}_{[k \neq i]} \exp(\tilde{z}_i . \tilde{z}_k / \tau)}. \tag{4}$$

By computing probabilities for both $SL_t$ and $SL_{t-1}$, we can write time distillation loss as:

$$\mathcal{L}_{TD}(\varphi_t; \varphi_{t-1}, \tau', \tau'') = \sum_{i=1}^{2N} -p(\tilde{x}_i; \varphi_{t-1}, \tau') . \log p(\tilde{x}_i; \varphi_t, \tau''), \tag{5}$$

where $\tau'$ and $\tau''$ represent the distillation-specific temperatures for the previous model and the current model, respectively.

**Knowledge Transfer from Reference:** The reference network encounters numerous unsupervised samples throughout its training and is expected to learn a rich representation space using the objective introduced in Section 3.1. This representation is used as guidance for the learner network, and the knowledge can be transferred to the learner network using an IRD loss similar to the Eq. 5:

$$\mathcal{L}_{KD}(\varphi_t; \theta_t, \tau', \tau'') = \sum_{i=1}^{2N} -p(\tilde{x}_i; \theta_t, \tau') . \log p(\tilde{x}_i; \varphi_t, \tau''). \tag{6}$$

This distillation is applied to the learner network based on the samples in $\mathcal{T}_t \cup \mathcal{M} \cup \hat{\mathcal{U}}_t$. It is noteworthy that this distillation, rather than using all of the unsupervised samples in $\mathcal{U}_t$, only uses the unsupervised samples $\hat{\mathcal{U}}_t$, which seems to be related to the training of the learner network.

### 3.4 THE URSL ALGORITHM

In summary, the model receives two sets of samples at each time step: $\mathcal{T}_t$ and $\mathcal{U}_t$. The reference network is trained on $\mathcal{U}_t$ using the self-supervised loss function introduced in Eq. 1. Then, an OoD detection and a pseudo-labeling technique introduced in Section 3.2, are used to segregate unsupervised samples in $\mathcal{U}_t$. Finally, the learner network is trained based on the weighted aggregation of three loss functions introduced in Section 3.3 by defining $\gamma$ and $\lambda$ as hyperparameters:

$$\mathcal{L}_s(\varphi_t) = \mathcal{L}_{\mathrm{sup}}(\varphi_t; \tau) + \gamma\mathcal{L}_{TD}(\varphi_t; \varphi_{t-1}, \tau', \tau'') + \lambda\mathcal{L}_{KD}(\varphi_t; \theta_t, \tau', \tau''). \tag{7}$$

---

**Algorithm 1** URSL: Unsupervised Reference and Supervised Learner

---

**Require:** A supervised dataset $\mathcal{D}_{sup} = \{\mathcal{T}_t\}_{t=1}^T$ and an unsupervised dataset $\mathcal{D}_{unsup} = \{\mathcal{U}_t\}_{t=1}^T$
 1: initialize $UR_0$ and $SL_0$ respectively with random parameters $\theta_0$ and $\varphi_0$
 2: **for** $t = 1, ..., T$ **do**
 3:      Initialize $\theta_t \leftarrow \theta_{t-1}$ and $\varphi_t \leftarrow \varphi_{t-1}$
 4:      Update $\theta_t$ based on $\mathcal{U}_t$ to minimize $\mathcal{L}_{unsup}(\theta_t; \tau)$ (Eq. 1)
 5:      Extract $\mathcal{P}^t$ using $\mathcal{T}_t \cup \mathcal{M}$ (Eq. 2)
 6:      Compute $S_l^t$ from $\mathcal{T}_t \cup \mathcal{M}$ (Section. 3.2)
 7:      Compute $\tau_{id} \leftarrow \mathrm{mean}\left(S_l^t\right) + \eta_{id}\mathrm{var}\left(S_l^t\right)$ and $\tau_{pl} \leftarrow \mathrm{mean}\left(S_l^t\right) + \eta_{pl}\mathrm{var}\left(S_l^t\right)$
 8:      Prepare $\hat{\mathcal{T}}_t$ and $\hat{\mathcal{U}}_t$ based on $\tau_{id}$, $\tau_{pl}$, and scores of $\mathcal{U}_t$ (Section 3.2)
 9:      **while** not done **do**
10:          Sample a batch $B$ from $\mathcal{T}_t \cup \mathcal{M} \cup \hat{\mathcal{T}}_t$
11:          Compute $\mathcal{L}_s \leftarrow \mathcal{L}_{\mathrm{sup}}(\varphi_t; \tau)$ based on $B$ (Eq. 3)
12:          **if** $t > 1$ **then**
13:              Update $\mathcal{L}_s \leftarrow \mathcal{L}_s + \gamma\mathcal{L}_{TD}(\varphi_t; \varphi_{t-1}, \tau', \tau'')$ based on $B$ (Eq. 5)
14:          Update $\mathcal{L}_s \leftarrow \mathcal{L}_s + \lambda\mathcal{L}_{KD}(\varphi_t; \theta_t, \tau', \tau'')$ based on a batch from $\mathcal{T}_t \cup \mathcal{M} \cup \hat{\mathcal{U}}_t$ (Eq. 6)
15:          Update $\varphi_t \leftarrow \varphi_t - \alpha\nabla_\varphi\mathcal{L}_s$
16:      Update $\mathcal{M}$ such that the number of samples for each class is the same.
17: Train the classifier head using $\mathcal{T}_T \cup \mathcal{M}$

---

## 4 EXPERIMENTS

**Benchmark Scenario:** To demonstrate the effectiveness of our method, we have performed several experiments in this section. We use two datasets for each experiment: the main and the peripheral. A small portion of the main dataset, which is determined by $P$, is selected as supervised data, the rest is considered as related unsupervised data, and all samples of the peripheral dataset are considered as (probably) unrelated unlabeled data. At each time step, 9000 examples from each unsupervised dataset are randomly sampled, shuffled together, and fed into the model as unsupervised data. In Appendix F, we provide the results of experiments in which the number of datasets inside $\mathcal{U}_t$ is greater than two datasets, and the environment is even more realistic.

The hyperparameters of our model are not dependent on the experiment configuration, and a general and consistent solution for all conditions is provided. We conducted a wide range of experiments to demonstrate the model's robustness in various scenarios. In our experiments, we used the CIFAR10, CIFAR100 (Krizhevsky et al., 2009), and Tiny-ImageNet (Le & Yang, 2015) datasets as the main or peripheral datasets, which are commonly used datasets in the open-set semi-supervised learning literature (Chen et al., 2020b; Huang et al., 2021; Yu et al., 2020); moreover, the settings of our experiments are known as the "cross dataset" setting in the open-set semi-supervised literature (Chen et al., 2020b). We have utilized ResNet-18 architecture as the backbone of both networks with a two-layer MLP on its head as the projector. The input images for the model are 32 x 32 pixels in size. Additionally, we use the notation $|\mathcal{M}|$ to show the size of the supervised memory introduced in Section 3. Further experimental setups and details are provided in Appendix B.

**Baselines: Co$^2$L** (Cha et al., 2021) can be seen as a simplified version of URSL in which there is no reference network and no means for using unsupervised samples. Therefore, we propose a modified version of Co$^2$L, **Co$^2$L-j**, in which the model is trained jointly by employing both a supervised and

Table 1: Accuracy of different models on the CIFAR10 dataset.

| Setting | Unsupervised | Method | | | | | |
|---|---|---|---|---|---|---|---|
| | Dataset | $Co^2L$ | $Co^2 2L\text{-}j$ | $Co^2L\text{-}p$ | GD | DM | URSL |
| $P = 0.01$ | CIFAR100 | $26.5_{\pm 1.6}$ | $36.0_{\pm 3.2}$ | $46.6_{\pm 0.3}$ | $24.0_{\pm 1.3}$ | $30.1_{\pm 0.7}$ | $\mathbf{58.2_{\pm 0.8}}$ |
| $|\mathcal{M}| = 50$ | Tiny-Imagenet | $26.5_{\pm 1.6}$ | $33.0_{\pm 1.5}$ | $42.7_{\pm 0.2}$ | $24.3_{\pm 1.4}$ | $29.7_{\pm 6.1}$ | $\mathbf{51.0_{\pm 11.9}}$ |
| $P = 0.1$ | CIFAR100 | $58.3_{\pm 1.0}$ | $52.7_{\pm 1.6}$ | $62.4_{\pm 0.1}$ | $48.1_{\pm 1.3}$ | $59.6_{\pm 4.2}$ | $\mathbf{72.8_{\pm 0.9}}$ |
| $|\mathcal{M}| = 200$ | Tiny-Imagenet | $58.3_{\pm 1.0}$ | $42.2_{\pm 2.1}$ | $61.3_{\pm 0.3}$ | $47.3_{\pm 1.4}$ | $42.1_{\pm 9.4}$ | $\mathbf{72.8_{\pm 0.6}}$ |
| | | $Co^2L$ | | GEM | | iCaRL | |
| $P = 1$ $|\mathcal{M}| = 200$ | None | $69.5_{\pm 0.6}$ | | $29.2_{\pm 0.5}$ | | $49.9_{\pm 1.7}$ | |

Table 2: Accuracy of different models on the CIFAR100 dataset.

| Setting | Unsupervised | Method | | | | | |
|---|---|---|---|---|---|---|---|
| | Dataset | $Co^2L$ | $Co^2L\text{-}j$ | $Co^2L\text{-}p$ | GD | DM | URSL |
| $P = 0.05$ | CIFAR10 | $15.9_{\pm 0.2}$ | $20.4_{\pm 0.4}$ | $21.0_{\pm 0.2}$ | $11.8_{\pm 0.9}$ | $24.2_{\pm 0.9}$ | $\mathbf{30.4_{\pm 0.2}}$ |
| $|\mathcal{M}| = 500$ | Tiny-Imagenet | $15.9_{\pm 0.2}$ | $16.9_{\pm 0.3}$ | $21.5_{\pm 0.2}$ | $11.5_{\pm 0.9}$ | $28.1_{\pm 1.2}$ | $\mathbf{30.5_{\pm 0.5}}$ |
| $P = 0.1$ | CIFAR10 | $25.1_{\pm 0.1}$ | $26.9_{\pm 0.4}$ | $28.3_{\pm 0.4}$ | $16.7_{\pm 1.1}$ | $33.6_{\pm 1.0}$ | $\mathbf{37.5_{\pm 0.4}}$ |
| $|\mathcal{M}| = 1000$ | Tiny-Imagenet | $25.1_{\pm 0.1}$ | $28.4_{\pm 1.4}$ | $28.9_{\pm 0.3}$ | $15.9_{\pm 0.2}$ | $\mathbf{38.5_{\pm 0.7}}$ | $37.2_{\pm 0.3}$ |
| | | $Co^2L$ | | GEM | | iCaRL | |
| $P = 1$ $|\mathcal{M}| = 1000$ | None | $35.1_{\pm 0.3}$ | | $22.4_{\pm 4.5}$ | | $34.4_{\pm 0.8}$ | |

an unsupervised contrastive loss on the supervised and unsupervised data, respectively. In another baseline, **$Co^2L\text{-}p$**, we only pre-train the model with unsupervised data available in the first time step and ignore the unsupervised data in the subsequent steps to avoid possible conflict with the supervised loss during continual learning. There are also two other baselines in the prior works that seem consistent with the OSSCL setting due to the presence of an OoD detection module. **GD** (Lee et al., 2019) trained an OoD module to recognize unlabeled data from previous classes among the entire unlabeled dataset. This in-distribution data was only used to combat catastrophic forgetting. **DM** (Smith et al., 2021) mainly changed GD setting through defining some policies over unlabeled data by using superclasses of the CIFAR100 and using the FixMatch method (Sohn et al., 2020). On the other side, we also report results of fully supervised continual learning for two popular continual learning models, **GEM** and **iCaRL**, and also the state-of-the-art **$Co^2L$**. These methods have access to all samples of the related dataset as labeled ones during continual learning but cannot use unlabeled samples from any source.

## 4.1 RESULTS

Tables 1, 2, and 3 show the classification accuracy at the final time step when the main datasets are selected as CIFAR10, CIFAR100, and Tiny-ImageNet, respectively. In almost all the experiments, URSL outperforms all other baselines. There are two reasons for the superiority of URSL over GD and DM: (i) Unlike GD and DM, which train OoD detection with a small number of labeled samples, OoD detection of URSL is based on the representation of the reference network, which is trained with a large amount of unlabeled data and has high discrimination power. (ii) GD only uses these unlabeled data to solve the forgetting, while URSL uses those to transfer a rich representation from the reference network to the learner network. Although $Co^2L\text{-}p$ and $Co^2L\text{-}j$ improved $Co^2L$, URSL outperformed them in all scenarios, showing the effectiveness of the proposed ideas compared with

Table 3: Accuracy of different models on the Tiny-Imagenet dataset.

| Setting | Unsupervised Dataset | Method | | | | | |
|---|---|---|---|---|---|---|---|
| | | $Co^2L$ | $Co^2L$-j | $Co^2L$-p | GD | DM | URSL |
| $P = 0.05$ | CIFAR10 | $8.4_{\pm 0.1}$ | $11.0_{\pm 0.1}$ | $12.8_{\pm 0.2}$ | $4.54_{\pm 0.02}$ | $4.8_{\pm 0.1}$ | $\mathbf{17.2_{\pm 0.1}}$ |
| $|\mathcal{M}| = 1000$ | CIFAR100 | $8.4_{\pm 0.1}$ | $10.8_{\pm 0.8}$ | $12.9_{\pm 0.1}$ | $5.7_{\pm 0.4}$ | $4.4_{\pm 0.5}$ | $\mathbf{17.5_{\pm 0.2}}$ |
| $P = 0.1$ | CIFAR10 | $15.1_{\pm 0.7}$ | $17.6_{\pm 0.6}$ | $18.4_{\pm 0.2}$ | $7.5_{\pm 0.2}$ | $5.6_{\pm 0.2}$ | $\mathbf{21.9_{\pm 0.2}}$ |
| $|\mathcal{M}| = 2000$ | CIFAR100 | $15.0_{\pm 0.7}$ | $18.4_{\pm 0.7}$ | $18.6_{\pm 0.1}$ | $7.9_{\pm 0.1}$ | $5.4_{\pm 0.4}$ | $\mathbf{20.8_{\pm 1.0}}$ |
| | | $Co^2L$ | | GEM | | iCaRL | |
| $P = 1$ $|\mathcal{M}| = 2000$ | None | $22.5_{\pm 0.5}$ | | $17.4_{\pm 0.3}$ | | $18.2_{\pm 0.2}$ | |

Table 4: Ablation of Eq. 7 on CIFAR100 classification with CIFAR10 dataset as peripheral.

| Version | URSL w/o $\mathcal{L}_{\text{sup}}$ | URSL w/o $\mathcal{L}_{TD}$ | URSL w/o $\mathcal{L}_{KD}$ | Only $\mathcal{L}_{\text{sup}}$ | Only $\mathcal{L}_{KD}$ | URSL |
|---|---|---|---|---|---|---|
| Acc.(%) | $25.7_{\pm 0.2}$ | $28.4_{\pm 0.7}$ | $28.9_{\pm 0.4}$ | $19.1_{\pm 0.9}$ | $28.2_{\pm 0.9}$ | $\mathbf{30.4_{\pm 0.2}}$ |

the naive approaches for incorporating unlabeled data. Furthermore, URSL achieved comparable or even better results than state-of-the-art full-supervised CL methods. This phenomenon suggests that URSL can benefit from unsupervised samples to mitigate the forgetting of previous classes or to learn a general representation that is proper for learning the classes that will be observed in the future.

We provide multiple benchmarks in Appendix D to show the robustness and power of our method. For instance, *After* and *Before* scenarios, in which the unlabeled related samples are respectively restricted to the future and past classes of the main dataset, prove that our method has positive forward and backward transfer. In a *Non-I.I.D.* scenario, we examined our method in an environment in which only a fraction of the classes of the main dataset are present in $\mathcal{U}_t$ at each time step. Additionally, Appendix E indicates that our method can achieve remarkable performance even in situations in which the ratio of the number of related unsupervised samples to the number of unrelated unsupervised samples is very low.

## 4.2 Ablation Studies

In this section, we conducted experiments to demonstrate the contribution of different components of the model to the final performance. To that end, we have selected CIFAR100 as the main dataset, CIFAR10 as the peripheral dataset, $P = 0.05$, and $|\mathcal{M}| = 500$. Table 4 indicates the model's performance in the experiments created by ablations over the losses of the model presented in Eq. 7:

**Effect of $\mathcal{L}_{\text{sup}}$:** $\mathcal{L}_{\text{sup}}$ induces the representation of the learner network to discriminate between classes directly by using supervised contrastive loss and labels; therefore, As the results suggest, this loss is important and contributes to the performance of the model. Adding $\mathcal{L}_{\text{sup}}$ to the *URSL w/o $\mathcal{L}_{\text{sup}}$* version is increased the performance by 4.7%. Moreover, although $\mathcal{L}_{KD}$ provides great discrimination for the learner network and achieves 28.2% accuracy, adding $\mathcal{L}_{\text{sup}}$ to this version still enhances the performance.

**Effect of $\mathcal{L}_{TD}$:** The role of $\mathcal{L}_{TD}$ is to transfer previously learned knowledge and reduce forgetting. The performance of the model is increased from 19.1% to 28.9% only by adding $\mathcal{L}_{TD}$ to *Only $\mathcal{L}_{\text{sup}}$* version. Furthermore, Although $\mathcal{L}_{KD}$ reduces forgetting in another way, adding $\mathcal{L}_{TD}$ to *URSL without $\mathcal{L}_{TD}$* increased performance from 28.4% to 30.4%. All mentioned comparisons indicate that this loss can effectively help the model to avoid forgetting.

**Effect of $\mathcal{L}_{KD}$:** $\mathcal{L}_{KD}$ is a new way to utilize unlabeled samples of the environment. This loss intends to transfer the reference network's rich knowledge about the environment to the learner network. As

Table 5: Ablation of OoD on CIFAR100 classification with CIFAR10 dataset as peripheral.

| | Experiment | Variant 1 | Variant 2 | Variant 3 | Variant 4 |
|---|---|---|---|---|---|
| Data | Eqs. 3 and 5 | $\mathcal{T}_t \cup \mathcal{M}$ | $\mathcal{T}_t \cup \mathcal{M}$ | $\mathcal{T}_t \cup \mathcal{M} \cup \hat{\mathcal{T}}_t$ | $\mathcal{T}_t \cup \mathcal{M} \cup \hat{\mathcal{T}}_t$ |
| | Eq. 6 | $\mathcal{T}_t \cup \mathcal{M} \cup \mathcal{U}_t$ | $\mathcal{T}_t \cup \mathcal{M} \cup \hat{\mathcal{U}}_t$ | $\mathcal{T}_t \cup \mathcal{M} \cup \mathcal{U}_t$ | $\mathcal{T}_t \cup \mathcal{M} \cup \hat{\mathcal{U}}_t$ |
| | Acc.(%) | $20.7_{\pm 1.2}$ | $24.5_{\pm 0.4}$ | $29.2_{\pm 0.5}$ | $\mathbf{30.4_{\pm 0.2}}$ |

it is shown, using this loss alone to train the learner network achieves great performance. In addition, adding $\mathcal{L}_{KD}$ to *Only* $\mathcal{L}_{\text{sup}}$ increases performance from 19.1% to 28.4%. Although $\mathcal{L}_{KD}$ and $\mathcal{L}_{TD}$ both reduce forgetting in different ways and have overlap in their function, adding $\mathcal{L}_{KD}$ to *URSL without* $\mathcal{L}_{KD}$ still boosts the performance.

It is worth mentioning that performance reduction from *Only* $\mathcal{L}_{KD}$ version to *URSL without* $\mathcal{L}_{\text{sup}}$ version is because of equality of $\mathcal{L}_{KD}$ and $\mathcal{L}_{TD}$ coefficients, $\lambda$, and $\gamma$, respectively. The high ratio of $\frac{\gamma}{\lambda}$ prevents the model from learning new tasks and reduces the plasticity of the model.

The next study, provided in Table 5, indicates the importance of the segregation module in providing $\hat{\mathcal{T}}_t$ and $\hat{\mathcal{U}}_t$ to the model. In the below paragraphs, the results are discussed:

**Effect of $\hat{\mathcal{T}}_t$:** Due to the fact that there exists a variety of images that differed from the current classes among unlabeled data, we are not able to use all unlabeled data naively in Eqs. 3 and 5. Therefore, In experiments, we investigate the effect of adding $\hat{\mathcal{T}}_t$ to the data of Eqs. 3 and 5. The results indicate the importance of adding $\hat{\mathcal{T}}_t$ in preventing the overfitting caused by the limited number of data from the past classes. Remarkable boost in the performance of *Variant 1*, and *Variant 3* can be seen by comparing them with *Variant 2*, and *Variant 4*, respectively.

**Effect of $\hat{\mathcal{U}}_t$:** Eq. 6 is designed to transfer the rich representation of the reference network to the learner network. The results show that adding $\mathcal{U}_t$ to this loss naively and without segregation leads to the transfer of irrelevant knowledge to the learner network. That unrelated transferred knowledge to tasks prevents the model from learning a discriminative representation for the target tasks. Segregation of $\mathcal{U}_t$ boosts the performance of *Variant 1*, and *Variant 3* by 3.8% and 1.2%, respectively.

## 5    CONCLUSION

In this paper, we present OSSCL, a novel setting for continual learning which is more realistic than previously studied settings. The setting assumes that the agent has access to a large number of unsupervised data in the environment, some of which are relevant to tasks due to the similarity between surroundings and tasks. As a possible solution for this setting, we presented a novel model, consisting of a supervised learner and an unsupervised reference network, to effectively utilize both supervised and unsupervised samples. The learner network benefits from three loss functions; the supervised loss function, which is formed based on limited supervised samples and segregated unsupervised samples, knowledge distillation through time, and representational guidance from the reference network. URSL has outperformed other state-of-the-art continual learning models with a considerable margin. The experiments and ablation studies demonstrate the superiority of the model and the effectiveness of each of its components.

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

Table 6: The Running time of a single task for the URSL and baselines for CIFAR100 classification with CIFAR10 as peripheral dataset

| Method | Co$^2$L-j | Co$^2$L-p | | GD | DM | URSL | |
|---|---|---|---|---|---|---|---|
| | | Pretrain (only once) | learning task | | | Reference | Learner |
| Training time(minutes) | 12 | 63 | 6.5 | 8.5 | 14.5 | 15.75 | 12 |

## A  RELATED WORKS

**Continual Learning**   Three families of approaches have been proposed to address the issue of forgetting in continual learning. Replay-based methods reuse samples from previous tasks either by keeping raw samples in a limited memory buffer (Rebuffi et al., 2017; Lopez-Paz & Ranzato, 2017; Aljundi et al., 2019) or by synthesizing pseudo-samples from past classes (Shin et al., 2017; Wu et al., 2018; van de Ven et al., 2020). Regularization-based methods aim to maintain the network's parameters stable across tasks by penalizing deviation from the important parameters for the previously learned tasks (Nguyen et al., 2018; Lee et al., 2017; Zenke et al., 2017; Cha et al., 2021; Rebuffi et al., 2017). Methods based on parameter isolation dedicate different parameters to each task by introducing new task-specific weights or masks (Rusu et al., 2016; Yoon et al., 2018; Wortsman et al., 2020).

Methods based on parameter isolation suffer either from extensive resource usage or capacity shortage when the number of tasks is large. Regularization-based methods are promising when the number of tasks is small; however, as the number of tasks increases, they become more prone to catastrophic forgetting and failure. However, replay-based methods have shown promising results in general continual learning settings. This work can be categorized as a replay-based method.

**Self-supervised Learning**   Self-supervised learning methods are being explored to learn a representation using unlabeled data such that the learned representation will be able to convey meaningful semantic or structural information. Based on this, various ideas, such as distortion (Alexey et al., 2015; Gidaris et al., 2018), jigsaw puzzles (Noroozi & Favaro, 2016), colorization (Zhang et al., 2016), and generative modeling (Vincent et al., 2008), have been investigated. Meanwhile, contrastive learning has played a significant role in recent developments of self-supervised representation learning. Contrastive learning involves learning an embedding space in which samples (e.g., crops) from the same instance (e.g., an image) are pulled together, and samples from different instances are pushed apart. Early work in this field incorporated some form of instance-level classification with contrastive learning and was successful in some cases. The results of recent methods such as SimCLR (Chen et al., 2020a), SwAV (Caron et al., 2020), and BYOL (Grill et al., 2020) are comparable to those produced by the state-of-the-art supervised methods.

**Knowledge Distilation**   Knowledge distillation aims to transfer knowledge from a teacher model to a student model without losing too much generalization power (Hinton et al., 2015). The idea was adapted in continual learning tasks to alleviate catastrophic forgetting by keeping the network's responses to the samples from the old tasks unchanged while updating it with new training samples (Shmelkov et al., 2017; Rebuffi et al., 2017; Li & Hoiem, 2016). iCaRL (Rebuffi et al., 2017) applies a distillation loss to maintain the probability vector of the last model outputs in learning new tasks, while UCIR (Hou et al., 2019) maximizes the cosine similarity between the embedded features of the last model and the current model. Co$^2$L (Cha et al., 2021) proposed a novel instance-wise relation distillation loss for continual learning that maintain features' relation between batch samples in the representation space.

**Semi-supervised learning**   In practical scenarios, the number of labeled data is limited; therefore, training the models using such limited labeled data leads to low performance. Due to this fact, semi-supervised learning methods try to utilize the unlabeled data among the labeled data to achieve better performance. There are three main categories of semi-supervised training methods: generative, consistency regularization, and pseudo-labeling methods. A generative method can learn implicit and

Table 7: hyperparameters search space

| Parameters | Values |
|---|---|
| $E_S$ | $\{100, 200\}$ |
| $E_{T_1}$ | $\{200, 300, 400\}$ |
| $E_{T_{>1}}$ | $\{100, 200\}$ |
| $\tau$ | $\{0.1, 0.5\}$ |
| $\lambda$ | $\{0.05, 0.2, 0.4\}$ |
| $(\eta_{id}, \eta_{pl})$ | $\{(-4, -2), (-2, 0), (-2, 2)\}$ |
| Optimizer | $\{$SGD + momentum, Adam$\}$ |
| Initial Learning rate | $\{0.1, 0.01\}$ |

transferable features of data in order to model data distributions more accurately in supervised tasks (Springenberg, 2015; Dumoulin et al., 2016; Li et al., 2017). Consistency regularization describes a category of methods in which the model's prediction should not change significantly if a realistic perturbation is applied to the unlabeled data samples (Rasmus et al., 2015; Laine & Aila, 2016; Tarvainen & Valpola, 2017). By pseudo-labeling, a trained model on the labeled set is utilized to provide pseudo-labels for a portion of unlabeled data in order to produce additional training examples that can be used as labeled samples in the training data set (Lee et al., 2013; Xie et al., 2020b; Pham et al., 2021). UDA (Xie et al., 2020a) and FixMatch (Sohn et al., 2020) are two examples of recent brilliant works in semi-supervised learning. UDA (Xie et al., 2020a) employs data augmentation methods as perturbations for consistency training and encourages the consistency between predictions on the original and augmented unsupervised samples. In the FixMatch method (Sohn et al., 2020), consistency regularization and pseudo-labeling are combined, and cross-entropy loss is used to calculate both supervised and unsupervised losses.

**Open-set Semi-supervised Learning**   Most semi-supervised learning methods assume that labeled and unlabeled data share the same label space. Nevertheless, in the Open-set Semi-supervised Learning setting, unlabeled data can contain categories that aren't present in the labeled data, i.e., outliers, which can adversely affect the performance of SSL algorithms. In UASD (Chen et al., 2020b), soft targets are produced by averaging predictions from some temporally ensembled networks, and out-of-distribution samples are detected using a simple threshold applied to the largest prediction score. Using a cross-modal matching strategy, Huang et al. (2021) trained a network to predict whether a data sample matches a one-hot class label or not. By using this module, they filter out samples that have low matching scores with all possible class labels. In Saito et al. (2021), inlier confidence scores were calculated using one-vs-all (OVA) classifiers. Furthermore, a soft-consistency regularization loss is also applied to enhance the OVA-classifier's smoothness, thereby improving outlier detection.

**Out-of-Distribution Detection**   Previous works in semi-supervised settings utilized an out-of-distribution detector in order to filter relevant unlabeled data. Some methods train a K-way classifier, assign pseudo-labels to unlabeled data and incorporate them in the training procedure. Due to the neural networks' overconfidence over even noisy data (Hsu et al., 2020), these methods use specific techniques to alleviate this phenomenon. Lee et al. (2019) trained a classifier using the confidence calibration technique in order to lower confidence over unseen data. In this work, they sampled a bunch of random data from a massive dataset like ImageNet and applied a loss to reduce the model confidence on them. Smith et al. (2021) used another technique called DeConf, by which they calibrated probabilities only using in-distribution data and without needing out-of-distribution data. In another type of method called "Learning from Positive and Unlabeled Data" (Comité et al., 1999; Elkan & Noto, 2008; Garg et al., 2021), authors train a binary classifier that demonstrates whether each input is in-distribution or not. Garg et al. (2021) proposed an iterative two-stage method in which first they estimate $\alpha$ that determines the mixture proportion of positive data among unlabeled data. Then, they train a classifier using estimated $\alpha$. They iterated these two stages until a convergence criterion was satisfied.

Table 8: Chosen hyperparameters for URSL

| Parameters | Values |
|---|---|
| $E_S$ | 200 |
| $E_{T_1}$ | 400 |
| $E_{T_{>1}}$ | 100 |
| Batch size | 512 |
| $\tau$ | 0.1 |
| $\tau'$ | 0.01 |
| $\tau''$ | 0.2 |
| $\lambda$ | 0.2 |
| $\gamma$ | 0.2 |
| $(\eta_{id}, \eta_{pl})$ | (-4, -2) |
| Optimizer | Adam |
| Initial Learning rate | 0.01 |
| Minimum learning rate | 1e-4 |

## B    DETAILS OF EXPERIMENTAL SETUPS

### B.1    DATASET DETAILS

The **CIFAR10** dataset consists of 60000 32x32 color images in 10 classes, with 6000 images per class. There are 50000 training images and 10000 test images. If this dataset is used as the main continual dataset, we randomly split it into 5 tasks with 2 classes per task. For this dataset, we used the ratios of $P = 0.01$ and $P = 0.1$ for the supervised samples, respectively equal to 50 and 500 samples per class.

The **CIFAR100** dataset contains 100 classes with 500 training and 100 test samples for each class. Each supervised task includes the training samples of 10 classes if this dataset is used as the main continual dataset. In Table 2 of the main paper, we used $P = 0.05$ and $P = 0.1$ configurations for this dataset, corresponding to 25 and 50 training samples per class.

**Tiny-Imagenet** is a subset of the Imagenet dataset which contains 200 classes, 100000 training samples, and 10000 test samples. Before using the dataset, we downsize the input images from 64x64 to 32x32 in order to make all image sizes equal. We split the dataset into 10 equally sized supervised tasks. Similar to CIFAR100, this dataset is used with $P = 0.05$ and $P = 0.1$ ratios which is equivalent to 25 and 50 training samples per class.

**Caltech256** is an object recognition dataset that contains 30607 real-world images from 257 categories. Images sizes are different from each other, and the minimum number of images per category is 80 images. We only use this dataset in Appendix F to increase the number of datasets in $\mathcal{U}_t$ in order to diversify the objects in unlabeled samples and provide a more realistic environment.

### B.2    TRAINING DETAILS

As explained in the main paper, we used two datasets for each experiment: (i) the main dataset to construct the supervised and related unsupervised samples and (ii) the peripheral dataset to provide the unrelated unsupervised samples. At each time step, 9000 unlabeled samples are provided from the main dataset and 9000 unlabeled samples from the peripheral dataset.

In our experiments, the ResNet-18 architecture is used as the encoder for our method, as well as all other baselines. In our method, starting from random initialization, the reference network is trained for $E_{T_1} = 400$ epochs at time step $t = 1$ to converge to a good representation. However, for subsequent time steps, it would only be trained for $E_{T_{>1}} = 100$ epochs. On the other hand, the learner network is trained for $E_S = 200$ epochs in all time steps like all the baseline methods whose main number of epochs is 200. The mean and standard deviation of results are obtained over 3 runs.

In Table 6, the required running times to train a single epoch of different models are reported. These results are recorded on a GeForce RTX 3080 Ti GPU.

Table 9: After and Before Benchmarks of CIFAR100 classification with the CIFAR10 dataset as peripheral

| Setting | Only Supervised | After | Before |
|---|---|---|---|
| Acc.(%) | $15.9_{\pm 0.2}$ | $19.1_{\pm 0.8}$ | $20.0_{\pm 0.1}$ |

Table 10: Only Related and Only Unrelated Benchmarks of CIFAR100 classification with CIFAR10 dataset as peripheral

| Setting | Only Supervised | Only Unrelated | Only Related | OSSCL |
|---|---|---|---|---|
| Acc.(%) | $15.9_{\pm 0.2}$ | $19.6_{\pm 0.5}$ | $29.4_{\pm 0.5}$ | $30.4_{\pm 0.2}$ |

### B.3 TUNING THE HYPERPARAMETERS

We created a validation set for all three main datasets by selecting 10% of the training samples at random and then performed a hyperparameter search according to Table 7. Table 8 shows the chosen hyperparameters obtained either by considering the validation results or by adapting from the $Co^2L$ paper. The strength of our proposed method is that the selected hyperparameters are invariant across different scenarios, and we used a single configuration for all experiments. For $Co^2L$, DM, and GD, we used the optimal hyperparameters if the authors reported it in the original papers. In addition, $Co^2L$-j and $Co^2L$-p used a similar set of hyperparameters as URSL except for the new hyperparameter introduced in $Co^2L$-j, where the unsupervised loss coefficient was set to 1.

### B.4 AUGMENTATIONS

To increase the diversity of training samples, following previous works (Cha et al., 2021; Chen et al., 2020a), we used the following augmentation techniques for all data:

1. **RandomResizedCrop**: The image is randomly cropped with the scale in $[0.2, 1]$ and then the cropped image will be resized to $32 \times 32$.

2. **RandomHorizontalFlip**: Each image is flipped horizontally with a probability $p = 0.5$, independently from other samples.

3. **ColorJitter:** The brightness, contrast, saturation, and hue of each image are changed with a probability of $p = 0.8$, with maximum strength $[0.4, 0.4, 0.4, 0.1]$, respectively.

4. **RandomGrayscale:** Images are converted to grayscale with probability $p = 0.2$.

### B.5 TRAINING CLASSIFIER

At the end of the training, we trained a linear classifier on the learner network's encoder head for 100 epochs using all memory data and the last time step labeled data $\mathcal{T}_T \cup \mathcal{M}$. We used Weighted Random Sampler to draw mini-batches due to class imbalance in labeled data.

## C THE PERFORMANCE OF THE OoD DETECTION

In this section, we evaluate the performance of the OoD detection module in two scenarios. For the first one, the number of related and unrelated data is 9,000 each, and for the other one, this number is 4,500. Precision and AUROC metric diagrams for the OoD detection module have been shown in those two settings (during the time steps) in Figures 2 and 3. It can be seen that the performance of the OoD detection module is improved over time due to the fact that it sees more classes and can detect class boundaries more precisely.

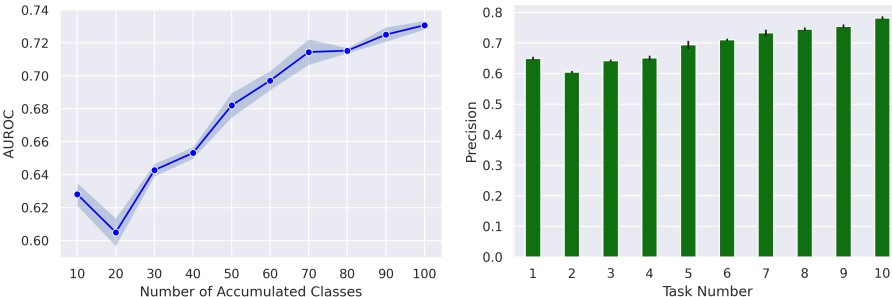

Figure 2: (left) AUROC of OoD detection based on the number of the main dataset seen classes for CIFAR100 classification with CIFAR10 dataset as peripheral when the number of related and unrelated data are 9000 (right) The precision of OoD detection at each task of CIFAR100 classification with the CIFAR10 dataset as peripheral when the number of related and unrelated data is 9000.

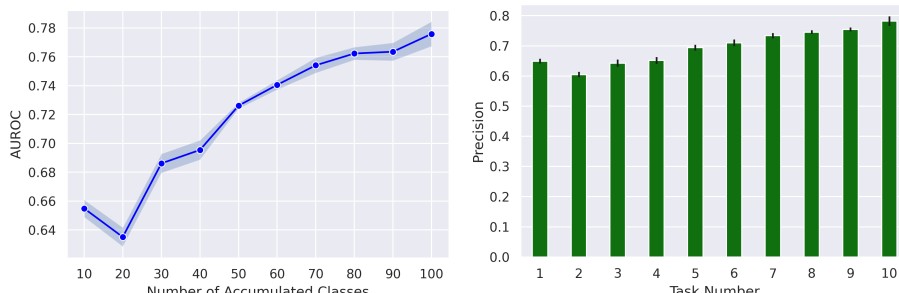

Figure 3: (left) AUROC of OoD detection based on the number of the main dataset seen classes for CIFAR100 classification with CIFAR10 dataset as peripheral when the number of related and unrelated data are 4500 (right) The precision of OoD detection at each task of CIFAR100 classification with the CIFAR10 dataset as peripheral when the number of related and unrelated data is 4500

## D    OTHER BENCHMARKS

In the "Other Benchmarks" section of the paper, we investigated different configurations to demonstrate the effectiveness and robustness of the URSL model in dealing with various conditions. All benchmarks are clarified, and the results are analyzed further below.

**After and Before**    In these two benchmarks, we assume that there are no unrelated samples among unlabeled data. The difference between these two settings is the presence of classes from the main dataset in the unlabeled data at any time step. More specifically, in the *After* scenario, unrelated samples are only from classes of the current time step and future classes from subsequent time steps are provided. However, in the *Before* scenario, the unlabeled data from only previous classes are presented. This experiment was designed to show that the URSL model can benefit from a positive forward/backward knowledge transfer from the unsupervised samples to the supervised tasks. As shown in Table9, in the *Before* scenario, it seems that visiting unlabeled data of previous classes helps to mitigate catastrophic forgetting (positive backward transfer). In contrast, in the *After* scenario, the model learns a decent representation space which is beneficial for learning the new coming classes (positive forward transfer).

**Only Related and Only Unrelated**    To investigate the effect of each type of unlabeled data on the model's functionality, we defined *Only Related* and *Only Unrelated* settings. As their names suggest, in the former, all unlabeled data at each task only contains main dataset samples. In contrast, in the latter, all unlabeled data are only from the peripheral dataset. In Table 10, comparing *Only Unrelated* with *Only Supervised* shows that even unrelated samples improve performance by enriching the representation of the reference network and providing a pivot model that prevents the

Table 11: Non-I.I.D. Benchmarks of CIFAR100 classification with CIFAR10 dataset as peripheral

| Setting | Only Supervised | Non-I.I.D. (25 %) | Non-I.I.D. (50 %) | OSSCL |
|---------|-----------------|-------------------|-------------------|-------|
| Acc.(%) | $15.9_{\pm 0.2}$ | $24.9_{\pm 0.6}$ | $27.4_{\pm 0.7}$ | $30.4_{\pm 0.2}$ |

Table 12: The effect of the number of related and unrelated samples on CIFAR100 classification with CIFAR10 dataset as the peripheral dataset.

| Related-Unrelated (samples) | 1000-9000 | 4500-4500 | 4500-9000 | 9000-4500 | 9000-9000 |
|-----------------------------|-----------|-----------|-----------|-----------|-----------|
| Accuracy(%) | $21.0_{\pm 1.1}$ | $26.8_{\pm 0.5}$ | $27.0_{\pm 0.6}$ | $30.0_{\pm 0.7}$ | $30.4_{\pm 0.2}$ |

learner network from high accumulative changes during the continual learning process. Also, the *Only Related* scenario demonstrates the effectiveness of existing related samples among unlabeled data in performance, and when compared with the *OSSCL* setting, the effectiveness of unrelated samples can be understood. It is worth mentioning that although most of the improvement is due to incorporating related samples, accessing pure related unlabeled data is not usually a realistic assumption. Instead, they are among a huge stream of unlabeled data containing unrelated samples. Therefore, we considered both related and unrelated datasets as unlabeled samples (in the *OSSCL* setting) and showed that properly employing these datasets (in URSL) further improves the results compared with the *OnlyRelated* case.

**Non-I.I.D.** The OSSCL scenario considers an I.I.D. assumption on the related unsupervised samples available in the environment. To challenge this assumption, we introduced a new benchmark in which the related data is generated only from a portion of the supervised classes at each time step. For example, in the *Non-I.I.D. (50 %)* experiment, the related unsupervised dataset, only includes the samples from half of the supervised classes, which are randomly selected at each time step. As it is shown in Table 11, the URSL model still demonstrates a good performance even with this limited access to the related unlabeled samples.

## E    NUMBER OF RELATED AND UNRELATED SAMPLES

In this section, we investigated the effect of the number and ratio of the related and unrelated samples among unlabeled data. In contrast to other baselines, URSL is able to utilize unrelated unlabeled samples to boost final performance even with an imbalanced number of related and unrelated unlabeled sets. Table 12 shows that increasing unrelated samples improves results slightly while increasing related samples provides the model with more in-distribution samples to improve its performance and combat catastrophic forgetting.

## F    MORE COMPLICATED ENVIRONMENTS

In this section, we examine the performance of our model in even more realistic environments by conducting more experiments in scenarios in which the unlabeled data is comprised of multiple datasets. Table 13 shows the performance of experiments. Besides the datasets we used in the main experiments, we also used Caltech256 (Griffin et al., 2007) in our experiments. At each experiment, we add 9000 samples from each dataset sampled randomly to the $\mathcal{T}_t$. The results suggest that our model is robust to a variety of unlabeled data and performs well in more realistic scenarios in which the model is exposed to plenty of unlabeled samples that most of which are not related to its target tasks.

Table 13: The results of using multiple datasets in $\mathcal{U}_t$ to stimulate a more realistic environment.

| Dataset1 | Dataset2 | Dataset3 | Dataset4 | Acc.(%) |
|----------|----------|----------|----------|---------|
| CIFAR10 | CIFAR100 | —— | —— | $30.4_{\pm 0.2}$ |
| CIFAR10 | CIFAR100 | Tiny-Imagenet | —— | $30.4_{\pm 0.6}$ |
| CIFAR10 | CIFAR100 | Caltech256 | —— | $31.4_{\pm 0.3}$ |
| CIFAR10 | CIFAR100 | Tiny-Imagenet | Caltech256 | $31.6_{\pm 0.7}$ |

Table 14: effect of different architectures for the reference and the learner network on CIFAR100 classification with CIFAR10 dataset as the peripheral dataset.

| Reference (#parameters) | Learner (#parameters) | Accuracy(%) |
|-------------------------|-----------------------|-------------|
| ResNet-18 (11.1M) | ResNet-18 (11.1M) | $31.2_{\pm 1.0}$ |
| ResNet-34 (21.2M) | ResNet-18 (11.1M) | $31.4_{\pm 0.6}$ |
| ResNet-18 (11.1M) | WideResNet-40-2 (2.24M) | $30.7_{\pm 0.5}$ |
| ResNet-50 (23.5M) | WideResNet-28-10 (36.48M) | $31.1_{\pm 0.2}$ |

## G  THE REFERENCE AND THE LEARNER ARCHITECTURES

The authors of $Co^2L$ (Cha et al., 2021) used ResNet-18 as the feature extractor architecture of their model. Following this design choice, we used the same architecture for both the learner and reference networks as well as all other models and baselines in all experiments to ensure a fair comparison. In this section, we investigate the effect of changing the architecture for the learner and the reference networks as reported in Table 14. As expected, the model's performance slightly increases as the number of model parameters grows. Moreover, deep ResNet architectures compared with wide ResNet architectures achieved better performance.

It is noteworthy that although we used a batch size of 512 in all of our experiments in other sections, the experiments in this section are performed with a batch size of 128 to meet the memory limit requirement, in addition to providing a fair comparison.

## H  MEMORY BUFFER SELECTION ALGORITHM

Selecting the suitable samples to be stored in the memory is an active area of research in continual learning (Bang et al., 2021; Tiwari et al., 2022; Isele & Cosgun, 2018). However, the purpose of our research was not to focus on memory selection policies. Therefore, we have used a random policy as it is widely adopted in many CL works (Prabhu et al., 2020; Guo et al., 2020; Balaji et al., 2020).

It is noteworthy that the segregation of unlabeled data provides more diverse data than what exists in the memory buffer from past classes. Nevertheless, because the stored samples in the memory buffer play an important role in segregating the unsupervised samples we conducted experiments using different selection algorithms for memory buffer samples. In addition to the "random" selection method, we defined three other selection strategies:

- Low-confidence: select the data on which the model has low confidence
- High-confidence: select the data on which the model has high confidence
- Rainbow (Bang et al., 2021): select from all the ranges of confidence. This algorithm calculates a confidence score for each sample and sorts all scores; then, it selects some data by considering the presence of samples from all ranges of model confidence.

Table 15 shows the performance of all algorithms:

As can be seen, the "Random" selection algorithm outperforms both the "High-confidence" and "Low-confidence" selection strategies by a good margin. Moreover, the "Rainbow" achieves similar results as the "Random" strategy.

Table 15: effect of different algorithms for data selection for memory buffer on CIFAR100 classification with CIFAR10 dataset as the peripheral dataset.

| Algorithm | Low-confidence | high-confidence | Rainbow | Random |
|---|---|---|---|---|
| **Accuracy(%)** | $67.9_{\pm 0.9}\%$ | $69.7_{\pm 1.0}\%$ | $72.5_{\pm 0.6}\%$ | $\mathbf{72.8_{\pm 0.9}\%}$ |

Table 16: Comparison of URSL with URSL with full-pretraining

| Method | URSL | URSL with Pretrain |
|---|---|---|
| Acc.(%) | $30.4_{\pm 0.2}$ | $31.3_{\pm 0.7}$ |

## I  FURTHER EXPERIMENTS

### I.1  PRETRAINING THE REFERENCE NETWORK

The reference network is expected to gradually absorb unsupervised knowledge from the environment. In this section, we designed an experiment to show the success of the reference network in continually learning the unsupervised samples. In this experiment, the reference network is first pre-trained with all unsupervised samples before starting the learning of the first supervised task. Next, the reference network is frozen, and its parameters are maintained throughout the entire learning procedure. Other training details and learning mechanisms of the learner network are the same as the original URSL model.

Table 16 demonstrates that pretraining only slightly improves the URSL results. This suggests that the unsupervised samples available in the environment are sufficient for the reference network to learn a proper representation even if the data is observed continually.

### I.2  SELF-SUPERVISED METHOD

In whole experiments, we used NT-Xent loss (Sohn, 2016), a popular and straightforward contrastive loss, which is widely used in self-supervised learning literature (Chen et al., 2020a) and achieved remarkable performances for training the reference network. However, our model and algorithm perform well regardless of the self-supervised loss used to train the reference network. To demonstrate it, we compared the results of our model with experiments in which the reference network is trained by BYOL (Grill et al., 2020), a different self-supervised algorithm from NT-Xent. Tables 17 and 18 report the results of runs for two different scenarios.

The advantage of BYOL over SimCLR is that it does not need negative data in training. Indeed, in our experiments, datasets have less diverse samples than huge datasets such as ImageNet; therefore, we expect BYOL to perform better than SimCLR. The empirical results are also a confirmation of this point. However, BYOL needs more time to obtain comparable results.

## J  LIMITATIONS

In our method, there exist several limitations. First, we have to keep a minimum number of main dataset samples from each class in the memory buffer in order to create more precise prototypes; Second, due to the non-parallelism of the training phase of the teacher network and the student network, the time complexity of our method is higher than other methods and baselines. Furthermore, our model performs worse if the number of samples for each class becomes rigorously imbalanced.

There exist other limitations related to the Open-Set Semi-Supervised Continual Learning scenario. Although this configuration seems more realistic than the previous works in literature, there may still exist situations in which the assumptions of OSSCL do not hold. For example, an agent may have limited access to both related and unrelated unlabeled samples in the environment. This will

Table 17: Comparison between NT-Xent and BYOL performance on CIFAR10 classification with Tiny-Imagenet dataset as peripheral

| Main dataset | Peripheral dataset | SSL method | Accuracy (%) | Time cost (mins) |
|---|---|---|---|---|
| CIFAR10 | Tiny-Imagenet | SimCLR | $72.8_{\pm0.6}$% | **136 mins** |
| CIFAR10 | Tiny-Imagenet | BYOL | $\mathbf{73.1_{\pm1.0}}$% | 258 mins |

Table 18: Comparison between NT-Xent and BYOL performance on CIFAR100 classification with CIFAR10 dataset as peripheral

| Main dataset | Peripheral dataset | SSL method | Accuracy (%) | Time cost (mins) |
|---|---|---|---|---|
| CIFAR100 | CIFAR10 | SimCLR | $30.4_{\pm0.2}$% | **219 mins** |
| CIFAR100 | CIFAR10 | BYOL | $\mathbf{32.3_{\pm0.8}}$% | 445 mins |

lead to poor performance of our model since it is designed to perform in a situation where plenty of unsupervised data exists.

## K  CODE AND DATA AVAILABILITY

The source code to reproduce the results of this paper is attached to this document. In this repository, there exists a README file containing instructions and configuration details. Moreover, the licenses of the freely available datasets and used source codes are also available in the README file.

