# OpenReview forum: "A distinct unsupervised reference model from the environment helps continual learning"
_ICLR.cc/2023/Conference — Submitted to ICLR 2023_

### Official Review · Reviewer_GwGC · 2022-10-24

**Confidence:** 4
**Correctness:** 3
**Technical Novelty And Significance:** 2
**Empirical Novelty And Significance:** 3
**Recommendation:** 5

**Clarity, Quality, Novelty And Reproducibility:**

Writing is clear and well-written in general. The novelty of the proposed scenario is not clear. The proposed method has a limited novelty, as it is combination of existing methods, but their combination produces a good performance in their experiment. I believe this work is reproducible with the provided code.

**Details Of Ethics Concerns:**

Nothing special.

**Strength And Weaknesses:**

Strengths

+ The proposed scenario is interesting.

+ The proposed method outperforms other methods in most cases.

Weaknesses

- It is hard to say the authors introduce a new scenario on continual learning. To me, the authors are just re-branding the setting proposed in prior works. Please make a more thorough comparison with prior works, [Lee et al.] and [Smith et al.] on the scenario. The authors have not explicitly claimed novelty on the scenario, but writing in abstract and intro is somewhat misleading, if they did not mean to claim the novelty.

- In fact, I could not see much difference on the setting from the prior work from [Lee et al.]. Unless specifically focusing on few- or low-shot setting, limiting the number of training data per class would not be a concern in the real-world setting. Rather, the scalability of the method would be more interesting and important. Based on the experimental result on the first few tables, the scalability with respect to the dataset size is partially confirmed, but the largest setting is still too small; only 10% of CIFAR. However, based on the experimental result on Table 14, the proposed method is not scalable with respect to the model size.

- Similar to above, I could not find the reason why the number of unlabeled data should be limited to only 9k per "time step" (this is confusing, "time step" sounds like one iteration or even smaller unit. I personally think "stage" would be a better term). There are plenty of unlabeled data in the open world, and there is no reason to randomly sample only 9k of them. Indeed, based on Table 13, the proposed model seems not scalable much with respect to the number of unlabeled data.

- Ablation studies on the hyperparameters are required. For example, the choice of $\eta_{id}$ and $\eta_{pl}$ seems critical to the performance of the proposed method.

- Training time is different over methods. Are authors sure if all methods are converged or early-stopped to show their best performance?

- $L_{KD}$ takes $\theta_t$ as a fixed set of parameters but missed.

- FixMatch and other few papers appear multiple times in the reference section.

**Summary Of The Paper:**

This paper proposes Open-Set Semi-Supervised Continual Learning, where the experimental setting is essentially a class-incremental learning scenario with unlabeled data from both in- and out-of-distribution. The proposed method consists of two components: the unsupervised reference model learns with the SimCLR loss and in charge of training representation and thresholding unlabeled data, and the supervised model learns with the supervised contrastive loss, previous model, and the unsupervised reference model. Experimental results show the effectiveness of the proposed method in small image benchmarks.

**Summary Of The Review:**

I appreciate thorough experimental results, but the proposed scenario and the experimental setting do not seem really match, and ablation study on the hyperparameter choice is not enough. I think this is a good paper if they limit their scenario to low-shot learning. Please answer my concerns above.

**post-rebuttal**

After reading the rebuttal, I decided not to change my rating, as I cannot agree with the validity of the proposed scenario.

The authors claimed that "The difference between OSSCL and the setting of Lee et al. [1] is that their setting is fully supervised, which makes the setting unrealistic."

I don't specifically agree with this argument. Once you incorporate unlabeled dataset, there is no significant difference between fully-supervised and semi-supervised, as we have both labeled and unlabeled data in both cases.
The proposed setting assumes that a portion of unlabeled dataset is guaranteed to share the same data distribution with the labeled dataset, which is more unrealistic to me. Rather, exposing a large unlabeled dataset to each time step of continual learning is more realistic. [1] used 80M Tiny Images as an unlabeled dataset, which is the source of CIFAR datasets. This means that unlabeled data in their setting includes data sharing a similar inductive bias to CIFAR in a natural way.

The scenario of [2] makes sense in some cases, as they limited the setting of each time step to the conventional semi-supervised learning; as authors said in the rebuttal, "unlabeled data at all time steps are drawn from the same dataset as the labeled data had been drawn from" in [2]'s setting.

However, this paper distinguishes main-labeled as labeled dataset and a union of main-unlabeled and peripheral (OOD) datasets, but at last, we end up with having a pair of labeled and unlabeled dataset, which is essentially not different from [1].
But the size of both labeled and unlabeled dataset is significantly smaller than [1], so I think the experimental setting in this paper is somewhat toy-ish and seems not scalable.

Again, I could not find the reason why we need to use "only 10% of CIFAR samples" as a labeled dataset, to make a portion of unlabeled data to share the same data distribution with the labeled dataset.
This not only makes the setting unrealistic, but also toy-ish, as we have a very small number of 32x32 training data per time step.

Also, after reviewing the paper again, an additional comment on Table 3 is that, the overall performance is too low, so I do not think the comparison is so meaningful.

---

> ### Author Response · Authors · 2022-11-11
> **Responses to Reviewer GwGC**
>
> We thank the GwGC Reviewer for valuable feedback.
>
> &nbsp;
>
> **Concern #1: New Scenario**
>
> It is widely believed that labeled supervised samples are hard to collect in real-world applications, and this is the main reason that semi-supervised learning literature has been introduced previously.
>
> The difference between OSSCL and the setting of Lee et al. [1] is that their setting is fully supervised, which makes the setting unrealistic; however, our setting assumes that at each time step, we have access to only a small fraction of the labeled dataset.
> Moreover, the difference between OSSCL and the setting of Smith et al. [2] is that although they restricted their labeled dataset and provided a semi-supervised scenario, their unlabeled data at all time steps are drawn from the same dataset as the labeled data had been drawn from.
>
> By considering these aspects, our setting is more realistic than the existing ones.
>
> &nbsp;
>
> [1] Lee, K., Lee, K., Shin, J., & Lee, H. (2019). Overcoming catastrophic forgetting with unlabeled data in the wild. In Proceedings of the IEEE/CVF International Conference on Computer Vision (pp. 312-321).
>
> [2] Smith, J., Balloch, J., Hsu, Y. C., & Kira, Z. (2021, July). Memory-efficient semi-supervised continual learning: The world is its own replay buffer. In 2021 International Joint Conference on Neural Networks (IJCNN) (pp. 1-8). IEEE.
>
> &nbsp;
>
> &nbsp;
>
> **Concern #2: Architecture Scalability**
>
> Using only 10% of CIFAR samples means that we only have limited access to training labels, not the training images themselves. In fact, this configuration is a common scenario widely used in semi-supervised learning where we assume that the labeling procedure is a costly effort and we have limited access to supervised samples. This doesn’t mean that our model can not benefit from the 90% remaining samples, but instead, we use those samples as unsupervised data.
>
> The purpose of the experiments in Table 14 was to investigate the effect of different architectures rather than directly investigating the model size. As we stated in Appendix G, deep ResNets achieve better results than wide ResNets.
> Although increasing the number of parameters increases the performance slightly, we believe that we should not expect a meaningful increase in performance for such spatially small images in the CIFAR datasets.
>
> &nbsp;
>
> **Concern #3: Unlabeled data scalability**
>
> Thank you for your suggestion. Yes, the “stage” term is also proper. Indeed, we used the “time step” term because it is a common term in continual learning literature.
>
> There are two kinds of increases in unlabeled data: 1) an increase in related unlabeled data and 2) an increase in unrelated unlabeled data. An increase in related unlabeled data has much more scalability than an increase in unrelated unlabeled data. As you can see in Table. 12, increasing related unlabeled data from 4500 to 9000 boosts the performance by 3.4%. Due to the fact that unrelated unlabeled data is only used to enrich the representation, an increase in the number of them is less scalable. However, an obvious improvement can be seen in Table 13.
>
> &nbsp;
>
> **Concern #4: Hyperparameter search**
>
> As mentioned in Table. 7, we selected some different values for our hyper-parameters and searched over their performance by using a hold-out validation dataset. We reported the best values in Table. 8.
>
> &nbsp;
>
> **Concern #5: Training time**
>
> Yes. We ensured the convergence of all baselines to achieve their best performance.
>
> &nbsp;
>
> **Concerns #6 and #7**
>
> Thank you for your consideration. We have included your suggestions in the new version.
>
> &nbsp;
>
> &nbsp;
>
> Please let us know if you still have any concerns, and we will be happy to address them.

---

> ### Author Response · Authors · 2022-12-01
> **A Response to Reviewer GwGC**
>
> Thank you again for your thoughtful comments and constructive suggestions.
>
> We have provided thorough responses and additional experimental results. We would appreciate it if you could read our responses and update the scores if your concerns have been addressed.
>
> We are glad to discuss further any concerns that you find not fully addressed. Thank you.

---

### Official Review · Reviewer_aMML · 2022-10-24

**Confidence:** 4
**Correctness:** 3
**Technical Novelty And Significance:** 3
**Empirical Novelty And Significance:** 2
**Recommendation:** 5

**Clarity, Quality, Novelty And Reproducibility:**

The paper is clearly written. The idea of the dual network is not novel but the proposed OSSCL setting could be interesting.

**Strength And Weaknesses:**

Strength:

The paper is clearly written and easy to read. The experiment part is solid and the idea of using a reference network from the environment is interesting. Each part of the losses the author introduced in their model is discussed by ablation studies.

Weaknesses:

1, Many of the components in the network training, such as the designing of loss functions, are borrowed from previous studies, which makes the work incremental. The authors should highlight the novelty they made compared to previous studies, instead of creating a problem (designing a paradigm, OSSCL) which has not be investigate previously, but only studied by the authors themselve.

2, In the experiment, the main and the perioheral dataset do share some underlying common distribution, and hence the out-of-distribution samples are not truly out-of-distribution samples. The authors should justify this point.

3, Citations need to be checked. Some of the citations are duplicate.


**Summary Of The Paper:**

The authors proposed a novel setting for continual learning called OSSCL, which assumes that the agent has access to a large number of unsupervised data in the environment with some of which are relevant to tasks. To solve the proposed task, the author introduced a dual network, consisting of a supervised learner and an unsupercised reference network. The dual model can effectively utilize both supervised and unsupervised samples in the data. Different elements in the model are evaluated by ablation studies. The proposed model also outperform some other SOTA continual learning models.

**Summary Of The Review:**

The model itself is based on many previous studies but the performance seems good, compared to previous models run by the authors themselves.

---

> ### Author Response · Authors · 2022-11-11
> **Responses to Reviewer aMML**
>
>
> We thank the aMML Reviewer for valuable feedback.
>
> &nbsp;
>
>
> **Concern #1: Novelty**
>
> In this paper, we have not focused on designing new losses. Instead, our main goal was to introduce a new mechanism for the semi-supervised CL scenario. In fact, we believe that the novelty lies in the architecture of our model and the interaction between the learning modules.
>
> As we have also reported in our experiments, trivial extensions to current CL algorithms can not easily solve the OSSCL problem, and the superior performance of our model compared to prior works is the strength of our work.
>
> &nbsp;
>
> **Concern #2: Datasets**
>
> Thanks for your attention to this point. We agree with your statement about some overlap between datasets. However, these overlaps are negligible. To address meticulously and numerically, the match proportion between the Tinyimagenet and the CIFAR-100 datasets is 13.5% [1]; Moreover, by comparing datasets classes, there exists one class overlap between the CIAFR-10 and the Tinyimagenet datasets and also one class overlap between the CIFAR-10 and the CIFAR-100 datasets.
>
> &nbsp;
>
> [1] Chen, Y., X. Zhu, W. Li, and S. Gong. “Semi-Supervised Learning under Class Distribution Mismatch”. Proceedings of the AAAI Conference on Artificial Intelligence, vol. 34, no. 04, Apr. 2020, pp. 3569-76,
>
> &nbsp;
>
> &nbsp;
>
> **Concern #3: Repeated Citation**
>
> Thanks for your notice. We have fixed them in the rebuttal revision.
>
>
> &nbsp;
>
> &nbsp;
>
> Please let us know if you still have any concerns, and we will be happy to address them.

---

> ### Author Response · Authors · 2022-12-01
> **A Response to Reviewer aMML**
>
> Thank you again for your thoughtful comments and constructive suggestions.
>
> We have provided thorough responses and additional experimental results. We would appreciate it if you could read our responses and update the scores if your concerns have been addressed.
>
> We are glad to discuss further any concerns that you find not fully addressed. Thank you.

---

### Official Review · Reviewer_G7Pp · 2022-10-24

**Confidence:** 3
**Correctness:** 3
**Technical Novelty And Significance:** 3
**Empirical Novelty And Significance:** 1
**Recommendation:** 5

**Clarity, Quality, Novelty And Reproducibility:**

The paper is mostly clearly written and the overall quality of the paper is high. The proposed approach is interesting, original and effective.



**Strength And Weaknesses:**

Strengths:
- The paper is well organized and well motivated.
- The ability of continual learning methods to utilize unlabeled data is important and the proposed method solves the problem in an effective way. The proposed approach based on reference and learner networks is interesting.
- The experiments are well designed and the method achieves large improvements over existing methods. Moreover, ablation study shows that each of the components of the objective function is essential.

Weaknesses/questions:
- The method has many hyperparameters and systematic analysis of the robustness to their selection is missing. It is not clear how to select these parameters.
- Some parts of the work need to be better explained. All parameters defined in the equations need to be defined which is not currently the case. The intuition behind each part of the objective function should be presented.
- Are there any assumptions on the proportion of the in-distribution compared to out-of-distribution unlabeled samples? Can the performance be degraded if unlabeled data is very different from the data that the model was trained on?
- Is the model applicable to large-scale datasets such as ImageNet? How does it compare to baselines in terms of scalability?


**Summary Of The Paper:**

This paper presents a new continual learning algorithm that exploits unsupervised data in the environment. The model consists of two main parts: (i) reference network that learns knowledge from unlabeled data, and (ii) learner network for solving specific task of interest by utilizing labeled data. The reference network is trained using self-supervised loss function and then out-of-distribution detection technique is applied to segregate unlabeled data. The learner network is trained using three loss functions that transfer knowledge through time, reference and utilize supervised samples. The method is tested on CIFAR-10, CIFAR-100 and Tiny-ImageNet datasets. It is shown to outperform existing methods.


**Summary Of The Review:**

Overall, this a good paper and presents an interesting and effective method for continual learning by exploiting unlabeled data in a realistic setting in which unlabeled data can contain out-of-distribution samples. The method outperforms baselines by a large margin.

---

> ### Author Response · Authors · 2022-11-11
> **Responses to Reviewer G7Pp**
>
> We thank the G7Pp Reviewer for valuable feedback.
>
> &nbsp;
>
> **Concern #1: Robustness of URSL**
>
> In Appendix B.3, we described our approach to choosing the hyper-parameters. We held out a validation set and searched over different values for hyper-parameters. Table 7 shows different values for hyperparameters, among which we performed the search. Finally, we reported the best values in Table 8.
>
> We believe that the hyperparameters are robust because as opposed to other papers like Co$^2$l [1] that report specific parameters for each specific scenario and setting, our model uses the same hyperparameters for all different scenarios.
>
> &nbsp;
>
> [1] Cha, H., Lee, J., & Shin, J. (2021). Co2l: Contrastive continual learning. In Proceedings of the IEEE/CVF International Conference on Computer Vision (pp. 9516-9525).
>
> &nbsp;
>
> &nbsp;
>
> **Concern #2: Better Explanation**
>
> Thanks for your notice. We will do our best to address your concern in the camera-ready version.
>
> &nbsp;
>
> **Concern #3: The proportion of related data to unrelated data**
>
> For simplicity in our main experiments, we fixed this ratio to 1. However, in Table 12, we investigated the effect of this ratio. The results show the robustness and effectiveness of our model even in intense imbalance ratios (e.g., 1 to 9).
>
> &nbsp;
>
> **Concern #4: Scalability**
>
> To the best of our knowledge, the CL community has not yet focused on investigating problems with this considerable number of continual classes. In fact, we believe that the performance of all CL methods greatly degrades in this configuration. Therefore, although the methods can be compared, the range of their performance is very low in this case.
>
> &nbsp;
>
> &nbsp;
>
> Please let us know if you still have any concerns, and we will be happy to address them.

---

> ### Author Response · Authors · 2022-12-01
> **A Response to Reviewer G7Pp**
>
> Thank you again for your thoughtful comments and constructive suggestions.
>
> We have provided thorough responses and additional experimental results. We would appreciate it if you could read our responses and update the scores if your concerns have been addressed.
>
> We are glad to discuss further any concerns that you find not fully addressed. Thank you.

---

### Official Review · Reviewer_4qUo · 2022-10-25

**Confidence:** 4
**Correctness:** 4
**Technical Novelty And Significance:** 2
**Empirical Novelty And Significance:** 3
**Recommendation:** 5

**Clarity, Quality, Novelty And Reproducibility:**

This is a well-organized paper and the proposed approach is intuitive to follow. My concern is related to the novelty of the paper which seems incremental and based on previous work heavily. The authors provide different experiment settings that outperform other benchmarks.


**Strength And Weaknesses:**

Strength:
- The paper is well organized and well written.
- The proposed model performs well compared with STOA. The experiments are fair and well executed.

Weaknesses:
- The novelty of the proposed approach is limited, most of the contributions are based on previous work such as: the SimCLR (Chenet al., 2020a) loss function, and an instance-wise relation distillation (IRD) loss to transfer knowledge from the previous time step to the current model (Cha et al., 2021).
- In (Further Experiments section 1.1) the authors said “In this experiment, the reference network is first pre-trained with all unsupervised samples before starting the learning of the first supervised task. Next, the reference network is frozen, and its parameters are maintained throughout the entire learning procedure.” Does this apply to avoid catastrophic forgetting?
- How does increasing the number of stored samples in the memory affect the performance?
- It would be nice if the authors can discuss the transfer and ability to scale to streams with a hundred tasks for the proposed model.

**Summary Of The Paper:**

The authors propose the URSL model “Unsupervised Reference network and a Supervised Learner network“, which consists of two parts: 1) The general task-agnostic reference network, which is responsible for absorbing information from unsupervised data in the environment, and 2) the learner network, which is designed to capture knowledge from a few supervised samples while it is also guided by the reference network. They utilize contrastive representation learning as a unified approach for training both the reference and the learner networks, which allows information to flow between these networks. They also combine with knowledge distillation techniques applied in the representation space to take advantage of unsupervised samples.

**Summary Of The Review:**

This is an ok paper that I feel is not strong enough. As I mentioned, my concern is related to the novelty. I like that the authors provide extensive experiments with different settings. Some discussion could be explained in more detail such as: if the model overcomes catastrophic forgetting by nature “the way it is constructed”, and the transferability and scalability of the model to a large number of tasks.

---

> ### Author Response · Authors · 2022-11-11
> **Responses to Reviewer 4qUo**
>
> ***UPDATE (13 Nov.): We added the variances of the memory buffer size experiments.***
>
> &nbsp;
>
> We thank the 4qUo Reviewer for valuable feedback.
>
> &nbsp;
>
>
> **Concern #1: Novelty**
>
> In this paper, we have not focused on designing new losses. Instead, our main goal was to introduce a new mechanism for the semi-supervised CL scenario. In fact, we believe that the novelty lies in the architecture of our model and the interaction between the learning modules.
>
> As we have also reported in our experiments, trivial extensions to current CL algorithms can not easily solve the OSSCL problem, and the superior performance of our model compared to prior works is the strength of our work.
>
> &nbsp;
>
> **Concern #2: Pretraining the Reference network**
>
> Yes. We intended to find whether catastrophic forgetting occurs for the Reference network or not.
>
> &nbsp;
>
> **Concern #3: Size of memory buffer**
>
> The size of the memory buffer is a determining factor in the performance of all memory-based CL methods. In most CL methods, increasing the memory buffer size leads to higher performance. Our paper is not an exception, and the below table reflects it. However, notice that a very large memory size violates the assumptions in continual learning, especially when the memory size becomes comparable to the size of the supervised data.
>
> &nbsp;
>
> | The size of the Memory buffer |  500 |  1000 |  1500 |
> |:-----------------------------:|:----:|:-----:|:-----:|
> |          Accuracy(%)          | $30.4_{\scriptscriptstyle\pm0.2}$ | $35.6_{\scriptscriptstyle\pm0.3}$ | $37.9_{\scriptscriptstyle\pm0.4}$ |
>
> &nbsp;
>
> Setting: CIFAR100 is the main dataset. The labeled dataset fraction is 5%. CIFAR10 is the peripheral dataset.
>
> ~~Note: we will update the results (variances) after finishing the experiments.~~
>
> &nbsp;
>
> **Concern #4: Scalability**
>
> To the best of our knowledge, the CL community has not yet focused on investigating problems with this considerable number of continual classes. In fact, we believe that the performance of all CL methods greatly degrades in this configuration. Therefore, although the methods can be compared, the range of their performance is very low in this case.
>
> &nbsp;
>
> &nbsp;
>
> Please let us know if you still have any concerns, and we will be happy to address them.

---

> > ### Comment · Reviewer_4qUo · 2022-11-29
> > **Acknowledgement**
> >
> > Thanks for the clarifications, they were all helpful, but overall, I am confirming my initial rating.

---

> > > ### Author Response · Authors · 2022-12-01
> > > **A Response to Reviewer 4qUo**
> > >
> > > Thank you again for your thoughtful and constructive comments.
> > >
> > > We have provided thorough responses and additional experimental results.
> > >
> > > Is there still a not fully addressed concern?

---

### Official Review · Reviewer_HjYX · 2022-10-25

**Confidence:** 4
**Correctness:** 3
**Technical Novelty And Significance:** 2
**Empirical Novelty And Significance:** 2
**Recommendation:** 3

**Clarity, Quality, Novelty And Reproducibility:**

Clarity - Overall, the paper is well-written an clear

Quality - There are no major flaws in the paper’s research quality besides those listed above.

Novelty - I find the overall novelty of the paper to be somewhat low (see above comments)

Reproducibility - Hyperparameter for experiments and other experimental details are provided in Appendix B.  Barring some bias from the selection of supervised samples to be included in the experiments, I feel this work is reasonably reproducible.

**Strength And Weaknesses:**

Strengths
1. I feel the framing of the continual learning problem where many unsupervised instances are provided for each task is a practically useful framing of continual learning.
2. The ablation study is a necessary part of an evaluation for URSL as it is complex and includes a number of additional terms in the supervised learning objective.

Weaknesses:
1. Overall, the empirical results are not very compelling.  The proposed method is rather complex but simpler methods that use no unsupervised data are fairly competitive to the point where one method that uses no unsupervised data actually outperforms URSL.  It is hard to argue for the practical utility of URSL given the relatively low or nonexistent performance increase over some of the baselines.
2. I find the overall experimental set up is somewhat limited.  First, accuracy results are reported only for the final task.  In continual learning the goal is to learn as tasks are presented to the learner.  Accuracy of intermediate tasks is a relevant and important result to report in the main body of the paper.  Second, the continual learning task presented has many variants based on how much supervised/unsupervised data is given and what data specifically is given as unsupervised.  The authors explore only a limited set of these, so it is hard to really understand the breadth of applicability of URSL.  Even simplifying the experiments by synthetically including known task-relevant data at some percentage to the unsupervised data to confirm the selection criteria is finding that data to learn from would be enlightening.
3. URSL borrows considerably from prior work.  As far as I can tell the basic procedures used to train each model are taken from prior work (namely SimCLR and Co^2L).  The main contributions is the method for sampling task-relevant features for Co^2L for the supervised model and two additional objective terms for knowledge transfer.  Without much principled justification (though some intuitive justification is provided), I do not find this work particularly novel at a technical level.

A few additional comments:
1. This problem seems highly related to PU learning (Garg et al; NeurIPS 2021 for example).  It would be good to see it discussed even briefly.
2. I don’t understand the intuition behind how thresholds are set for selecting unsupervised instances.  The thresholds are set relative to a mean similarity.  This seems to result in cases where an entire batch could be dissimilar and still many instances can be selected.  Said another way, if one batch has a low mean similarity (many instances are dissimilar to the prototype) and another batch has high mean similarity (many instances are similar to the prototype), the same number of instances can be selected for both if their \eta parameters and variances are the same.  This seems counter-intuitive.
3. What does the following statement mean: “It is noteworthy that the representation space of the reference network is chosen for OoD detection since it provides better sample discrimination than any other representation space obtained by training over a small number of labeled samples.”? How can such a claim be justified?
4. Why two thresholds for selection?  Intuitively if an instance is deemed task-relevant, it should have a task-specific pseudo label, yes?

**Summary Of The Paper:**

This paper focuses on a version of a continual learning problem where for each continual task a learner is given a set of labeled instance, and a set of unlabeled instances in which some are relevant to the task and some are not.  The authors make the claim that this version of the continual learning task is most realistic in practice and is not explicitly studied in prior work. The authors propose a method called URSL that 1) Trains a "reference" network using a contrastive objective on unsupervised data 2) Uses the reference network to select task relevant instance from the unsupervised data 3) Trains a separate supervised learning model using labeled instances, instances from a memory buffer, and instances determined as task-relevant (but used as negative samples).  The authors perform a series of experiments where the supervised labels are from one data set and the unsupervised data is from an entirely different data set.  URSL performs slightly better than most baselines when it comes to accuracy of the model after the final task is learned, but in some cases performs worse.


**Summary Of The Review:**

While the authors propose to focus on perhaps the most realistic version of the continual learning problem, I feel that the proposed method is not strong in technical novelty and doesn’t introduce significant methodological or theoretic contributions, which makes empirical performance a focus.  Unfortunately, I also did not find the empirical results compelling by not performing consistently better than baselines where no unsupervised data is used.  For these reasons, I argue for rejection.

---

> ### Author Response · Authors · 2022-11-11
> **Responses to Reviewer HjYX**
>
> We thank the HjYX reviewer for valuable feedback.
>
> &nbsp;
>
> **Concern #1: Performance of URSL**
>
> The simpler methods, such as Co$^2$l, GEM, and iCaRL, that we reported in the main tables are fully supervised and assume access to “all” of the labeled data. Therefore, we consider their results like an upper-bound performance. In some cases, our method even outperforms those upper bounds. This shows the effectiveness of our model in using unsupervised data.
>
> &nbsp;
>
> **Concern #2: Experiments setup**
>
> We reported the accuracy of the “final model” on all tasks, not only the “final task”. In fact, we followed the metric commonly used by the CL community, and the reported accuracies are the accuracy over all classes measured using all the test data.
>
> In the main table of the paper, we decided to report the configuration that is mostly used by the semi-supervised learning community. In appendix E, however, we reported diverse experiments on different configurations to address your concern by considering different ratios of relevant to irrelevant samples (Please refer to Table 12). Even in scenarios where the ratio of relevant to irrelevant data is low (e.g., 1 to 9), our model shows a remarkable performance.
>
> &nbsp;
>
> **Concern #3: Novelty**
>
> In this paper, we have not focused on designing new losses. Instead, our main goal was to introduce a new mechanism for the semi-supervised CL scenario. In fact, we believe that the novelty lies in the architecture of our model and the interaction between the learning modules.
>
> As we have also reported in our experiments, trivial extensions to current CL algorithms can not easily solve the OSSCL problem, and the superior performance of our model compared to prior works is the strength of our work.
>
> &nbsp;
>
> **Comment #1: PU learning**
>
> Thank you for your suggestion. We found the topic really interesting and related to our work. Therefore we included the topic in our related work section, and we are planning to include the methods from PU learning in the next versions of our work.
>
> &nbsp;
>
> **Comment #2: OoD Thresholds**
>
> The unlabeled data selection is not batch-wise. In fact, at each time step, both thresholds are calculated using all supervised samples, and then they are fixed. This means that all of the unsupervised samples are processed using the same threshold. Please look at lines 5-8 in our algorithm in the paper.
>
> &nbsp;
>
> **Comment #3: The power of unlabeled representation**
>
> When designing a prototypical-based OoD module, we should choose a representation space on which we apply the OoD. To that end, we have two options: we could follow previous works [1, 2] and use the representation space trained on few supervised samples, or we can apply the OoD on the representation space of the Reference network, which was trained using a huge amount of unsupervised data. We selected the second approach because our intuition was that the second approach provides better discrimination power.
>
> We have tested this hypothesis empirically. In order to compare the discrimination power of two representations (one learned by unlabeled samples and one learned by some small fraction of labeled data), we trained a classifier on the top head of learned representations using all labeled data of the dataset. The test accuracy of the unlabeled learned representation significantly outperformed the other one.
>
> &nbsp;
>
> [1] Lee, K., Lee, K., Shin, J., & Lee, H. (2019). Overcoming catastrophic forgetting with unlabeled data in the wild. In Proceedings of the IEEE/CVF International Conference on Computer Vision (pp. 312-321).
>
> [2] Smith, J., Balloch, J., Hsu, Y. C., & Kira, Z. (2021, July). Memory-efficient semi-supervised continual learning: The world is its own replay buffer. In 2021 International Joint Conference on Neural Networks (IJCNN) (pp. 1-8). IEEE.
>
> &nbsp;
>
> **Comment #4: OoD different Thresholds**
>
> We designed the in-distribution threshold to be looser than the pseudo-label threshold because we believe that there may exist some unsupervised samples related to the task, but that data is not directly in the classes that are going to be solved. In fact, we call a training sample to be task-relevant if it has similar semantics to the target task we are solving, but it is not necessarily included in one of the classes of that task. Using a looser threshold for knowledge transfer can lead to better generalization by transferring the task-relevant information to the Learner network.
>
> &nbsp;
>
> &nbsp;
>
> Please let us know if you still have any concerns, and we will be happy to address them.

---

> > ### Comment · Reviewer_HjYX · 2022-11-22
> > **Response**
> >
> > Thank you for your rebuttal.  It helped me understand your position better.  However, my opinion is unchanged, and have decided to keep my score.

---

> > > ### Author Response · Authors · 2022-12-01
> > > **A Response to Reviewer HjYX**
> > >
> > > Thank you again for your insightful responses and comments. We are glad that our responses have partly addressed your concerns.
> > > Indeed, we thought that there was a misunderstanding about our paper based on your provided summary, #1 and #2 Weaknesses.
> > > As you decided to keep your score unchanged, we worried that our responses maybe not have correctly clarified the issue.
> > >
> > > Therefore, we again want to give a better explanation of these concerns.
> > >
> > > &nbsp;
> > >
> > > **Concern #1: The performance of URSL**
> > >
> > > You mentioned that "simpler methods that use no unsupervised data are fairly competitive to the point where one method that uses no unsupervised data actually outperforms URSL."
> > >
> > > However, the point here is that in real-world problems, having access to supervised data is restricted, and the number of supervised data is limited (This is why lots of research has been conducted on the "Semi-supervised learning" area). Therefore, labeled data is valuable, and it is trivial that we get better results using more labeled data.
> > >
> > > **However, a remarkable and interesting result is that, in some cases, URSL even outperformed supervised results.**
> > >
> > > &nbsp;
> > >
> > >
> > > **Concern #2: Experiments setup**
> > >
> > > In the Class incremental learning problem in CL, the overall accuracy at the final time step is the most important metric [1]. Moreover, we believe that Table 12 in the appendix can fully address your second concern.
> > >
> > > &nbsp;
> > >
> > > Van de Ven, G. M., & Tolias, A. S. (2019). Three scenarios for continual learning. arXiv preprint arXiv:1904.07734.
> > >
> > > &nbsp;
> > >
> > > **Concern #3: Novelty**
> > >
> > > Moreover, Are there any remained concerns about the novelty of our algorithm for the Semi-supervised continual learning scenario?
> > >
> > > &nbsp;
> > >
> > > &nbsp;
> > >
> > > Please let us know if you still have any concerns that have not been fully addressed, and we will be happy to address them.

---

### Decision · Program_Chairs · 2023-01-20

**Decision:**

Reject

**Justification For Why Not Higher Score:**

No reviewer recommends acceptance

**Justification For Why Not Lower Score:**

N/A

**Metareview: Summary, Strengths And Weaknesses:**

The paper focuses on a version of continual learning where for each task a learner is given a set of labeled and unlabeled instances, and proposes a method called URSL to tackle this problem. The strengths of the paper include its framing of the continual learning problem and the design of experiments. However, the paper has several weaknesses, including limited and unconvincing empirical results, a lack of justification for its technical novelty, and a lack of discussion of related work. Additionally, method for setting thresholds for selecting unsupervised instances, among others, is not well explained and need to be further clarified. Overall, all reviewers recommended rejection therefore it cannot be accepted at this time.